# Association between tumor size and peritumoral brain edema in patients with convexity and parasagittal meningiomas

Chansub Shin[1], Jae Min Kim[1], Jin Hwan Cheong[1], Je Il Ryu[1], Yu Deok Won[1], Yong Ko[2], Myung-Hoon Han[1]*

1 Department of Neurosurgery, Hanyang University Guri Hospital, Guri, Gyonggi-do, Korea, 2 Department of Neurosurgery, Hanyang University Medical Center, Seongdong-gu, Seoul, Korea

* gksmh80@gmail.com

## Abstract

### Background and purpose

Peritumoral brain edema (PTBE) is a common complication in meningioma and disruption of the tumor-brain barrier in meningioma is crucial for PTBE formation. To evaluate the association between meningioma size and PTBE, we measured meningioma volumes using the 3D slicer in patients with convexity and parasagittal meningiomas.

### Methods

Receiver operating characteristic curve analysis was used to determine the optimal cut-off meningioma volume values for predicting PTBE occurrence. Logistic regressions were used to estimate the odds ratios for PTBE occurrence in patients with convexity and parasagittal meningiomas according to several predictive factors.

### Results

A total of 205 convexity or parasagittal meningioma patients with no other brain disease who underwent one or more contrast-enhanced brain MRIs were enrolled in this 10-year analysis in two hospitals. The optimal cut-off meningioma volume value for prediction of PTBE in all study patients was 13.953 cc (sensitivity = 76.1%; specificity = 92.5%). If a meningioma is assumed to be a complete sphere, 13.953 cc is about 2.987 cm in diameter.

### Conclusions

Our study suggests a cut-off value of 3 cm meningioma diameter for prediction of PTBE in patients with convexity and parasagittal meningiomas. We believe that we have revealed why the meningioma diameter of 3 cm is clinically meaningful.

**Data Availability Statement:** All relevant data are within the paper.

**Funding:** This study was funded by the Basic Science Research Program through the National

Research Foundation of Korea (NRF) funded by the
Ministry of Science, ICT & Future Planning (NRF-
2019R1G1A1085289) and research funding from
Hanyang University (HY-201900000003370).

**Competing interests:** The authors have declared
that no competing interests exist.

## Introduction

Meningioma is the second most common intracranial tumor in adults and peritumoral brain
edema (PTBE) is a common complication in meningioma patients that causes significant mor-
bidity [1]. Previously, various risk factors have been reported which are associated with PTBE
formation in meningioma patients that include meningioma size, presence of the tumor-brain
barrier, location, tumor margin shape, hyperintensity on T2WI, and vascular endothelial
growth factor expression [1–4]. However, because meningioma is a tumor with a wide variety
of locations, most previous studies that investigated the association between meningioma and
PTBE had substantial tumor location heterogeneity. It has been reported that the meningioma
location affects PTBE occurrence and grade [3, 5, 6]. Therefore, we felt it was necessary to
include meningiomas with similar locations in this study to reduce the effect of location het-
erogeneity on the association between meningioma and PTBE.

Tumor-brain barrier disruption in meningioma is crucial for PTBE formation [1]. We
recently reported a possible association between PTBE and the brain-meningioma interface
and meningioma volume after radiation therapy [7]. In that study, because radiation may
aggravate the damaged tumor-brain contact interface, we hypothesized that greater damage to
the tumor-brain interface due to large tumor volume would be associated with a higher proba-
bility of tumor-brain barrier disruption leading to PTBE after radiation therapy [8]. However,
if a meningioma is not treated with surgery or radiation, tumor size may be a major factor
affecting the integrity of the tumor brain barrier. Therefore, we wanted to evaluate the pure
association between meningioma size and PTBE occurrence without surgery or radiation ther-
apy in patients with meningiomas at similar locations. In addition, based on clinical experience
or implicit consent, most physicians decided on whether to perform surgery or initiate radia-
tion treatment based on a meningioma diameter of 3 cm, without any scientific reference [9–
13]. Therefore, to verify the validity of this size as a clinically meaningful standard for treat-
ment, we needed to evaluate the cut-off value for meningioma size that predicts PTBE develop-
ment in patients with meningiomas.

To test this hypothesis, we measured meningioma volumes from contrast-enhanced brain MRI
using the 3D slicer tool in patients with convexity and parasagittal meningiomas who did not
undergo surgery or radiation therapy. We evaluated predictive factors for PTBE in those patients.

## Methods

### Study patients

We retrospectively investigated all consecutive patients who were diagnosed with intracranial
meningioma in the Department of Neurosurgery of our hospitals from January 1, 2009 to
December 31, 2018. We initially identified 2,170 patients with meningioma in both hospitals. All
meningiomas were diagnosed by radiologic findings alone or pathological confirmation after
surgical resection. To reduce the possible effect of location heterogeneity, we only included con-
vexity and parasagittal meningiomas in this study. We then excluded patients with other brain
diseases that included brain tumor other than meningioma, dementia, stroke (ischemic, hemor-
rhagic), traumatic brain injury, and brain infections. Patients without contrast-enhanced MRI
showing meningioma at least once were also excluded, because contrast-enhanced brain MRI is
necessary to measure meningioma volume more precisely using the 3D slicer and to assess radio-
logical predictive factors for PTBE in patients with convexity and parasagittal meningiomas. If
multiple MRIs were performed, the last follow-up MRI was used for the analysis. When the
patients had surgery or radiation therapy for meningioma, we included the follow-up MRI
images, which were taken immediately prior to surgery or radiation therapy.

This study was approved by the Institutional Review Boards of Hanyang University Medical Center in both Seoul and Guri and conformed with the tenets of the Declaration of Helsinki. Owing to the retrospective nature of the study, the requirement for informed consent was waived. All individual records were anonymized prior to analysis.

## Brain MRI acquisition and volumetric assessment

Ingenia and Achieva (Philips, Eindhoven, the Netherlands; Philips, Böblingen, Germany) 1.5 Tesla (2009–2010) and 3.0 Tesla (2011–2018) MRI scanners were used for image acquisition in all patients. Routine contrast enhanced brain MRI protocol was performed, including axial T1WI, T2WI, FLAIR, and gadolinium-enhanced T1-weighted images, with a slice thickness of 1.0–5.0 mm [14].

We measured meningioma volume with 3D slicer software using version 4.6.2, (http://www.slicer.org), and the reliability the 3D slicer has been described elsewhere [15, 16]. We previously reported several studies using 3D slicer [17–19]. All procedures were conducted by a skilled 3D-slicer user. The contrast-enhanced T1-weighted MR images were used for volumetric analysis in all patients. The stepwise methods of volumetric assessment for meningioma using the 3D slicer were as follows: (1) brain MRI DICOM files of the study patients from the picture archiving and communication system (PACS) were loaded to the 3D slicer software; (2) threshold-based methods were performed to segment the convexity or parasagittal meningioma; (3) the results were then manually refined to complete the fine segmentation; (4) Model Maker function was used to generate 3D reconstruction of meningioma; and (5) the Label Statistics function was finally used to estimate meningioma volume (Fig 1A and 1B).

## Radiographic variables

All meningiomas and PTBE were radiologically confirmed on brain MRI by the radiologists. According to a previous study, PTBE grade was classified as grade 1 (edematous area was less than the tumor volume), grade 2 (edematous area and tumor volumes were equal), and grade 3 (edematous area was larger than the tumor volume) [20]. The signal intensity of meningioma on T2WIs was categorized into low/iso signal intensity or high signal intensity relative to the signal intensity of the cortical gray matter on T2W brain MR images [21]. Irregular tumor margin was defined when lobulation of a tumor's shape was seen at the brain–tumor interface [22]. The peritumoral rim was defined as the presence of a CSF layer showing hypo-signal intensity on T1WI and hyper-signal intensity on T2WI in the brain-tumor interface [21]. All MR images were reviewed by an experienced investigator who was blinded to patient details.

## Statistical analysis

Patient data are expressed as mean ± standard deviation or median with interquartile range for continuous variables, and as a count and percentage for discrete variables. The chi-square test and Student's t-test were conducted to assess differences between the two groups. Younger patients were defined as aged <65 years, and older patients were defined as aged ≥65 years.

Receiver operating characteristic (ROC) curve analysis was used to determine the optimal cut-off meningioma volume values that predict PTBE occurrence. The meningioma volumes were used as the test variable and the existence of PTBE as the state variable (dependent variable) in the ROC curve analysis. We set the non-PTBE group as code 0 and PTBE grade 1, 2, and 3 as code 1 and input the state variable.

Box plots were used to visualize meningioma volume differences between the non-PTBE and PTBE groups and between younger and older age groups according to with or without PTBE and PTBE grades.

## A

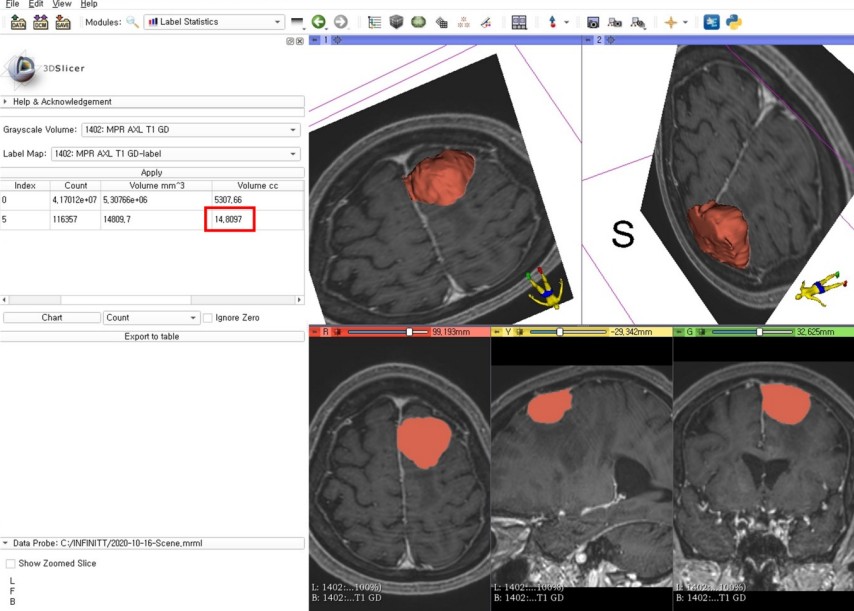

## B

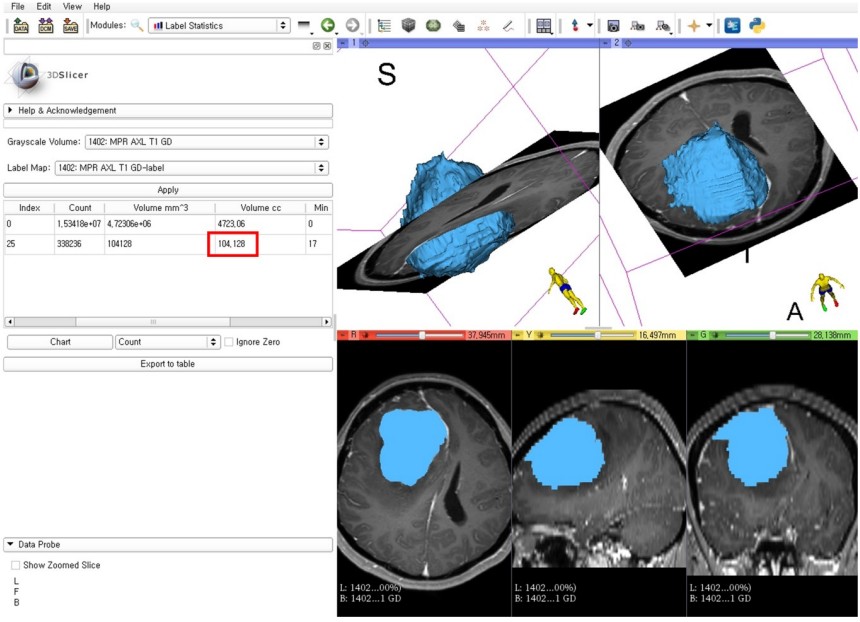

**Fig 1. Segmentation of meningioma with a 3D-reconstructed model using the 3D slicer and estimation of the tumor volume (red box indicates the volume).** (A) an example of a 3D-reconstructed meningioma from one of the study patients and the volume shows about 14.8cc; (B) an example of a 3D-reconstructed meningioma from one of the study patients and the volume shows about 104.1 cc.

A scatterplot with a line determined by locally weighted scatter plot smoothing (LOWESS) was performed to graphically represent the association between PTBE grades and meningioma volumes according to sex and age groups. The rationale and detailed methods underlying the use of LOWESS were previously reported [23].

Uni- and multivariate logistic regressions were used to estimate the odds ratios (ORs) with 95% confidence intervals (CIs) for the PTBE occurrence in patients with convexity and parasagittal meningiomas according to several predictive factors (sex, age, meningioma volume, meningioma pathologic grade, tumor signal intensity on T2WI, tumor margin, and peritumoral rim). Missing values among the meningioma pathologic grade variable were coded with "99" and included in the statistical analysis [24].

A p value < 0.05 was considered statistically significant. All statistical analyses were performed using R software version 3.6.3 and SPSS for Windows, version 24.0 software (IBM, Chicago, IL).

## Results

### Characteristics of study patients

Altogether, 205 convexity or parasagittal meningioma patients (>18 years of age) with no other brain disease who underwent one or more contrast-enhanced brain MRIs were enrolled in this 10-year study at two university hospitals. A total of 71 (34.6%) meningioma patients showed PTBE. The mean patient age was 65.8 years, and 21.5% of patients were men. There were significant differences in age, meningioma volume, tumor signal intensity on T2WI, tumor margin, and peritumoral rim between the non-PTBE and PTBE groups. Detailed information for the study patients is presented in Table 1.

### Determination of the optimal meningioma volume for predicting PTBE

Fig 2A shows significantly larger meningioma volumes in PTBE patients compared to non-PTBE patients (p<0.001).

When patients were divided into age groups, there was no significant difference in meningioma volume between the age groups in the PTBE group (p = 0.209) (Fig 2B). The optimal meningioma volume cut-off value for prediction of PTBE in all study patients was 13.953 cc (AUC [Area Under the Curve] = 0.916; sensitivity = 76.1%; specificity = 92.5%; p<0.001) (Fig 2C). If meningioma is assumed to be a complete sphere, 13.953 cc is about 2.987 cm in diameter. When we classified patients by age group, the optimal meningioma volume cut-off values for prediction of PTBE were 14.082 cc (AUC = 0.923; sensitivity = 80.0%; specificity = 93.9%; p<0.001) in younger patients and 9.174 cc (AUC = 0.911; sensitivity = 82.6%; specificity = 86.8%; p<0.001) in older patients (Fig 2D). Again, assuming meningioma is a complete sphere, 14.082 cc is about 2.996 cm in diameter and 9.174 cc is about 2.597 cm in diameter.

### Association between PTBE grade and meningioma volume

The LOWESS curves showed similar meningioma volumes between PTBE grades (Fig 3A).

When the patients were classified by sex, they also showed similar patterns of association between PTBE grades and meningioma volumes (Fig 3B). However, although low sample size may bias the results (n = 16), there were significant differences in meningioma volumes between age groups for PTBE grade 2 (p = 0.044) (Fig 3C and 3D). This suggests that elderly patients are more vulnerable to higher PTBE grade for the same or smaller meningioma volume compared to younger patients.

### Independent predictive factors for PTBE occurrence in convexity and parasagittal meningiomas

The results of uni- and multivariate logistic regression analyses are shown in Table 2.

**Table 1. Characteristics of patients with convexity and parasagittal meningiomas classified according to peritumoral edema.**

| Characteristics | Peritumoral edema (-) | Peritumoral edema (+) | Total | p |
|---|---|---|---|---|
| Number (%) | 134 (65.4) | 71 (34.6) | 205 | |
| Sex, male, n (%) | 27 (20.1) | 17 (23.9) | 44 (21.5) | 0.529 |
| Age, mean ± SD, y | 64.3 ± 13.3 | 68.5 ± 11.4 | 65.8 ± 12.8 | 0.026 |
| Time interval between diagnosis and the follow up MRI that was used for the study, mean ± SD, day | 399.8 ± 907.8 | 209.3 ± 604.7 | 333.8 ± 819.1 | 0.113 |
| Peritumoral edema grade, n (%) | | | | |
| Grade I | N/A | 34 (47.9) | 34 (16.6) | |
| Grade II | N/A | 16 (22.5) | 16 (7.8) | |
| Grade III | N/A | 21 (29.6) | 21 (10.2) | |
| Meningioma location, n (%) | | | | 0.697 |
| Convexity | 83 (61.9) | 42 (59.2) | 125 (61.0) | |
| Parasagittal | 51 (38.1) | 29 (40.8) | 80 (39.0) | |
| Meningioma volume, mean ± SD, cc | 5.6 ± 8.1 | 36.5 ± 30.5 | 16.3 ± 24.0 | <0.001 |
| Meningioma volume, median (IQR), cc | 2.6 (1.5–6.0) | 28.5 (14.1–55.7) | 5.5 (2.1–18.9) | <0.001 |
| Surgery, n (%) | 29 (21.6) | 50 (70.4) | 79 (38.5) | <0.001 |
| Pathologic classification, n (%) | | | | 0.891 |
| WHO grade I | 23 (17.2) | 39 (54.9) | 62 (30.2) | |
| WHO grade II/III | 6 (4.5) | 11 (15.5) | 17 (8.3) | |
| N/A | 105 (78.4) | 21 (29.6) | 126 (61.5) | |
| Pathology among patients who underwent surgery, n (%) | | | | 0.617 |
| Meningothelial | 7 (24.1) | 9 (18.0) | 16 (20.3) | |
| Fibrous | 5 (17.2) | 14 (28.0) | 19 (24.1) | |
| Transitional | 9 (31.0) | 11 (22.0) | 20 (25.3) | |
| Angiomatous | 0 | 3 (6.0) | 3 (3.8) | |
| Psammomatous or microcystic | 2 (6.9) | 2 (4.0) | 4 (5.1) | |
| Atypical | 6 (20.7) | 10 (20.0) | 16 (20.3) | |
| Anaplastic | 0 | 1 (2.0) | 1 (1.3) | |
| Ki 67 among patients who underwent surgery, n (%) | | | | 0.439 |
| <1% | 13 (44.8) | 18 (36.0) | 31 (39.2) | |
| ≥1% | 16 (55.2) | 32 (64.0) | 48 (60.8) | |
| Tumor signal intensity on T2WI, n (%) | | | | 0.001 |
| Low/Iso | 107 (79.9) | 41 (57.7) | 148 (72.2) | |
| High | 27 (20.1) | 30 (42.3) | 57 (27.8) | |
| Tumor margin, n (%) | | | | <0.001 |
| Smooth | 67 (50.0) | 13 (18.3) | 80 (39.0) | |
| Irregular | 67 (50.0) | 58 (81.7) | 125 (61.0) | |
| Peritumoral rim, n (%) | | | | <0.001 |
| No | 36 (26.9) | 40 (56.3) | 76 (37.1) | |
| Yes | 98 (73.1) | 31 (43.7) | 129 (62.9) | |

SD, standard deviation; IQR, interquartile range; WHO, World Health Organization; N/A, not available; T2WI, T2-weighted imaging.

In the univariate analysis, age, meningioma volume, tumor signal intensity on T2WI, tumor margin, and peritumoral rim were significant PTBE predictors. However, multivariate logistic analysis showed that only meningioma volume was an independent PTBE predictive

factor in convexity and parasagittal meningioma patients (OR, 1.19; 95% CI, 1.05 to 1.35; p = 0.006; per 1 cc increase of meningioma volume).

## Discussion

Our study showed an approximately 1.2-fold increased risk for PTBE per 1 cc increase in meningioma volume in patients with convexity and parasagittal meningiomas. The overall meningioma volume threshold for predicting PTBE occurrence with high specificity and sensitivity was approximately 14 cc. When meningioma is assumed to be a complete sphere, an

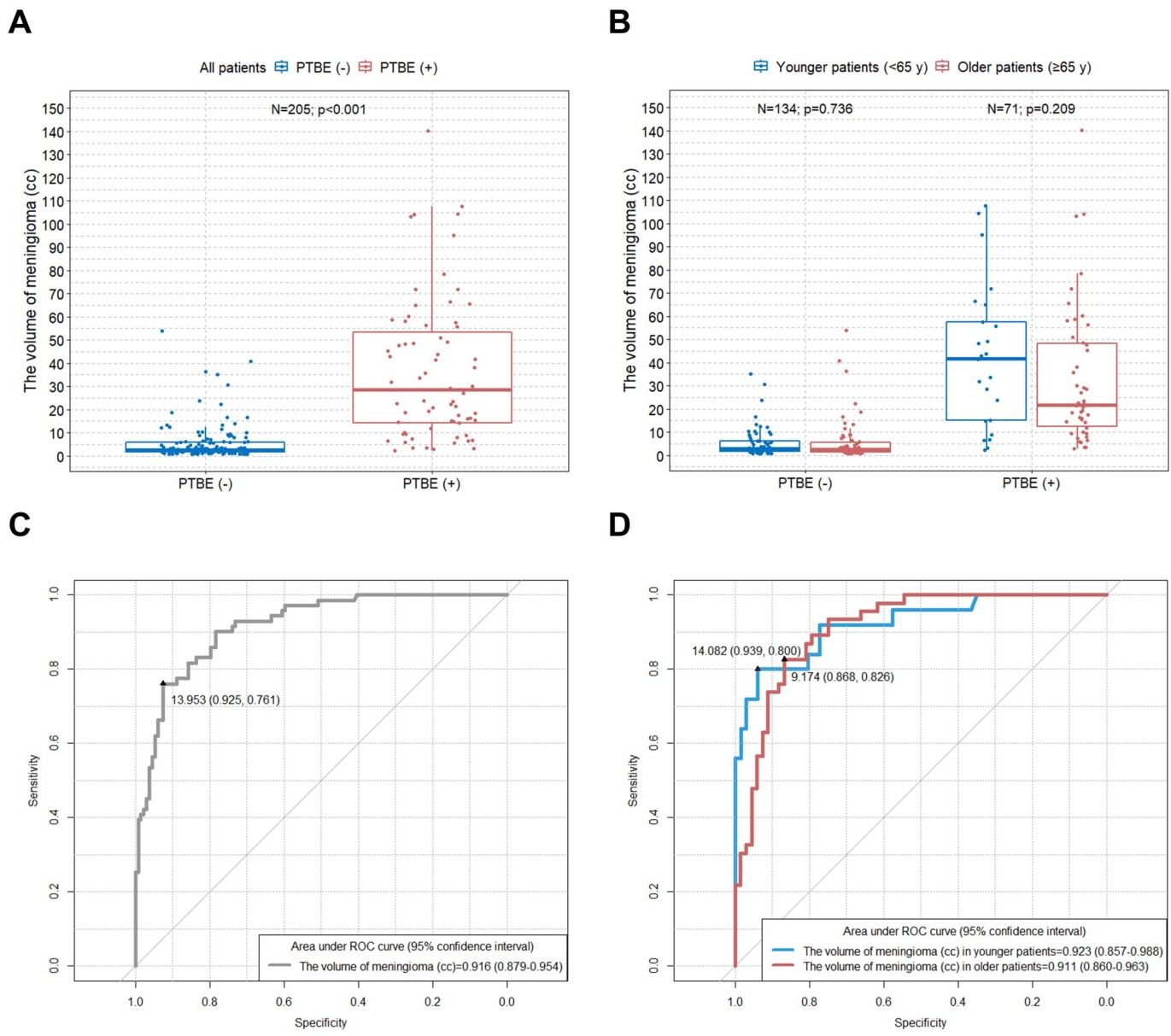

**Fig 2. Comparison of meningioma volume between PTBE and non-PTBE groups and determination of the optimal meningioma volume cut-off values for prediction of PTBE in patients with convexity and parasagittal meningiomas.** Boxplots of (A) meningioma volume in all patients and (B) meningioma volume classified by age group according to PTBE. ROC curve to identify the optimal cutoff values of (C) meningioma volume in all patients and (D) meningioma volume classified by age group for prediction of PTBE. PTBE = peritumoral brain edema; ROC = receiver operating characteristic.

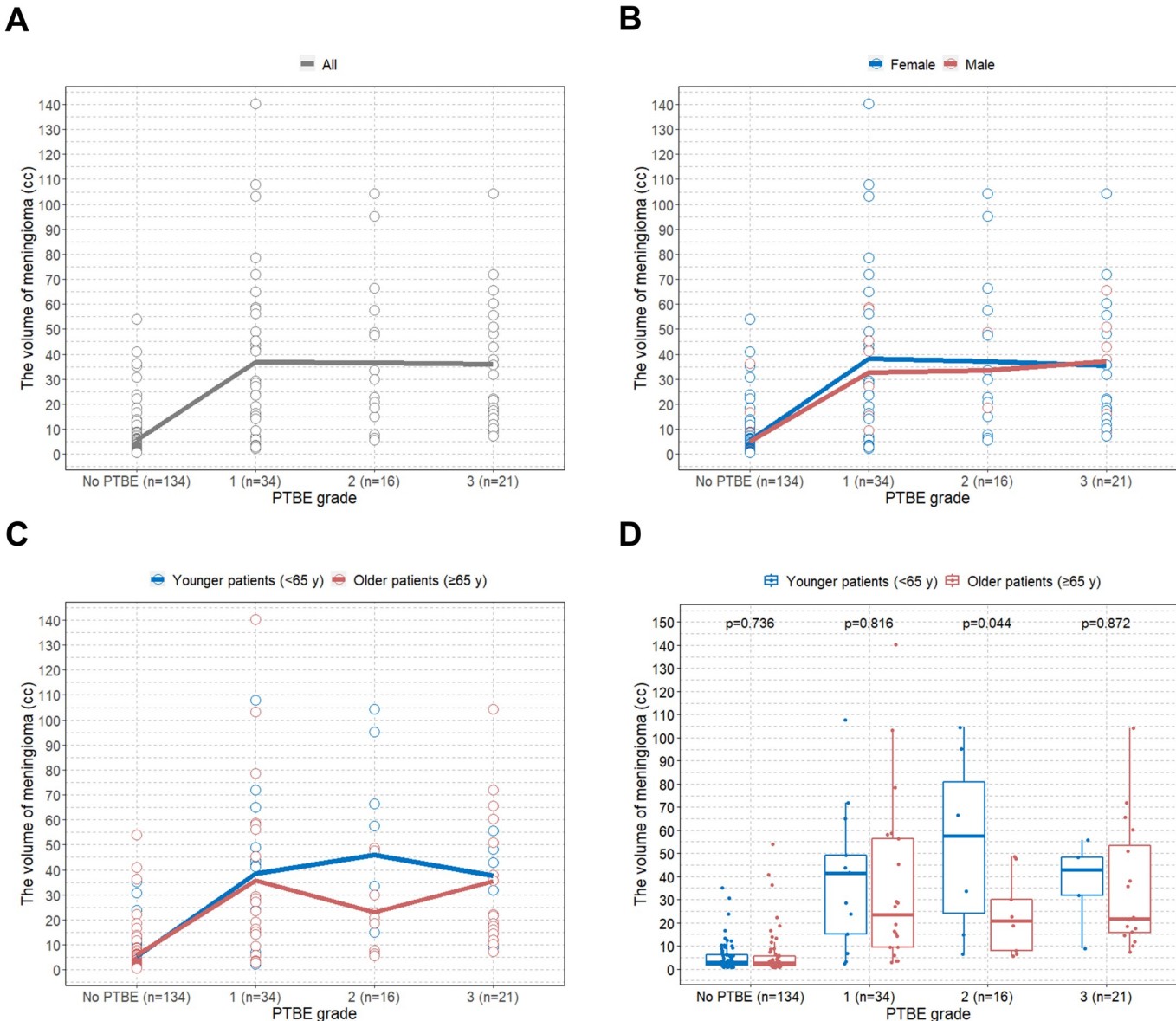

**Fig 3. Scatterplot with LOWESS curve and boxplots for assessment of the association between PTBE grade and meningioma volume.** (A) Scatterplot with LOWESS curve showing the association between PTBE grade and meningioma volume in all patients; (B) Scatterplot with LOWESS curve showing the association between PTBE grade and meningioma volume classified by sex; (C) Scatterplot with LOWESS curve showing the association between PTBE grade and meningioma volume classified by age group; (D) Boxplots showing the comparison of meningioma volume between age groups according to PTBE grade. LOWESS = locally weighted scatter plot smoothing; PTBE = peritumoral brain edema.

approximate 3 cm diameter was the optimal cut-off value for PTBE prediction. On the other hand, elderly patients were more vulnerable to both PTBE occurrence and higher grade PTBE for the same or smaller meningioma volume compared to younger patients.

To the best of our knowledge, this is the first study to suggest an optimal cut-off value for meningioma volume that predicts PTBE in patients with convexity and parasagittal meningiomas. Determining whether to treat small or large meningiomas with stereotactic radiosurgery or fractionated stereotactic radiotherapy or surgery based on a meningioma diameter of 3 cm without any specific reference studies or data has been accomplished by implicit consent

**Table 2. Univariate and multivariate logistic regression analyses of the association between peritumoral edema and various variables in patients with convexity and parasagittal meningiomas.**

| Variable | Univariate analysis | | Multivariate analysis | |
|---|---|---|---|---|
| | OR (95% CI) | p | OR (95% CI) | p |
| Sex | | | – | |
| Female | Reference | | Reference | |
| Male | 1.25 (0.63–2.49) | 0.529 | 2.39 (0.33–17.46) | 0.390 |
| Age (per 1-year increase) | 1.03 (1.00–1.05) | 0.028 | 1.04 (0.98–1.11) | 0.184 |
| Meningioma volume (per 1 cc increase) | 1.13 (1.09–1.17) | <0.001 | 1.19 (1.05–1.35) | 0.006 |
| Pathology | | | | |
| WHO grade I | Reference | | Reference | |
| WHO grade II/III | 1.08 (0.35–3.32) | 0.891 | 0.48 (0.04–5.56) | 0.558 |
| Tumor signal intensity on T2WI | | | | |
| Low/Iso | Reference | | Reference | |
| High | 2.90 (1.54–5.46) | 0.001 | 0.44 (0.05–4.05) | 0.471 |
| Tumor margin | | | | |
| Smooth | Reference | | Reference | |
| Irregular | 4.46 (2.24–8.90) | <0.001 | 4.09 (0.83–20.10) | 0.083 |
| Peritumoral rim | | | | |
| No | 3.51 (1.92–6.43) | <0.001 | 2.42 (0.52–11.35) | 0.261 |
| Yes | Reference | | Reference | |

OR, odds ratio; CI, confidence interval; WHO, world health organization

among clinicians [9–13]. Therefore, we believe that our findings may be valuable because we have shown why a meningioma 3 cm diameter can be established as a clinically meaningful standard for the treatment of convexity and parasagittal meningiomas.

It is well known that even small sized brain tumors such as glioblastoma or metastatic tumors usually cause PTBE. However, in contrast to those tumors, meningiomas are encapsulated by the arachnoid membrane and are separated from the underlying normal cerebral cortex. It has been reported that the arachnoid membrane may act as a mechanical and biochemical buffer that blocks the spread of vasogenic edema fluid and endothelial growth factor/vascular permeability factor from meningiomas to the white matter [8, 25]. Therefore, it is well accepted that the disruption of the brain-meningioma interface is a crucial component of PTBE formation [1, 8, 25]. A previous study regarding microscopic anatomy of the brain-meningioma interface showed that the brain-meningioma interface consisted of hyperplastic arachnoid trabeculae (shown in Fig 1A of the study) [26]. It is naturally assumed that as a tumor grows, the arachnoid trabeculae may be sandwiched between normal brain cortex and the meningioma. Therefore, the larger the tumor, the greater the probability of damage to the brain-meningioma interface including the arachnoid trabeculae which may lead to tumor-brain barrier disruption and PTBE formation. Most previous studies also agreed that meningioma volume is closely related to PTBE occurrence [2–4, 8]. We believe that our findings may additionally suggest that for convexity or parasagittal meningioma, when these tumors are assumed to be a complete sphere, a diameter of about 3 cm may be the possible cut-off point for disruption of the brain-meningioma interface.

On the other hand, in the elderly, a meningioma diameter of less than 3 cm was the cut-off value for prediction of PTBE in our study. Previously, it was revealed that arachnoid trabeculae and granulations are composed of type 1 collagen [27]. Based on the above concept, we have previously reported clinical studies that suggest a possible association between osteoporotic

conditions and arachnoid trabeculae integrity because bone and arachnoid trabeculae are composed of type 1 collagen [7, 17, 28]. Therefore, because osteoporosis is more common with increasing age, it is possible to postulate that the elderly may have a greater chance of having weakened arachnoid trabeculae integrity which constitutes the brain-meningioma interface compared to younger patients. This may increase the likelihood of tumor-brain barrier disruption leading to PTBE occurrence as tumor size increases in elderly patients with convexity and parasagittal meningiomas. Further, loosening of microstructural integrity and a volume reduction of white matter in the elderly may easily allow direct transmission of edematous fluids into the white matter. This may also increase the possibility of PTBE occurrence or even a more advanced grade of PTBE in the elderly compared to younger meningioma patients [29].

Our study has some limitations. First, due to its retrospective nature, inherent limitations exist. Second, technical errors may have been made in the measurement of meningioma volumes with the 3D slicer. Third, we only included convexity and parasagittal meningiomas and therefore our findings may not be applied to other types of meningioma. However, this also can be a strength of our study because the above inclusion criteria may reduce the possible effect of meningioma location heterogeneity on the association between meningioma volume and PTBE occurrence. In addition, convexity and parasagittal locations are the most common types of meningioma among intracranial meningiomas and symptomatic PTBE is a common complication after radiosurgery for convexity and parasagittal meningiomas [30–32]. Therefore, we believe that our study which included only convexity and parasagittal meningiomas, may help clinicians understand the underlying mechanisms of PTBE occurrence in the context of meningioma. Fourth, not all meningiomas were pathologically confirmed by surgical treatment. Therefore, there may be bias in our results. Fifth, the meningioma diameter of 3 cm derived from the volume threshold of 13.953 cc in our study was solely based on the assumption that it had a spherical shape. Our premise is that the closer the meningioma shape is to a sphere, the more accurately PTBE prediction can be based on a diameter of 3 cm. In contrast, PTBE prediction for irregularly shaped convexity or parasagittal meningiomas should be based on the direct volume threshold instead of a diameter of 3 cm. Lastly, there may be other factors that should be considered in PTBE development in meningiomas, such as tumor location, histological differentiation, hormonal receptors, and arterial tumor supply, which are not covered in our study [33–35].

In conclusion, despite these limitations, our study is the first to suggest a cut-off meningioma volume for prediction of PTBE in patients with convexity and parasagittal meningiomas. The meningioma volume threshold for predicting PTBE occurrence was approximately 14 cc. In addition, we believe that we have revealed why the meningioma diameter of 3 cm is clinically meaningful. Although not statistically significant, we believe that our findings may suggest that PTBE occurs more frequently in older meningioma patients than it does in younger patients. However, further studies are needed to confirm these initial findings. We expect our findings may enhance understanding of the association between meningioma size and PTBE formation.

## Author Contributions

**Conceptualization:** Myung-Hoon Han.

**Data curation:** Chansub Shin.

**Formal analysis:** Myung-Hoon Han.

**Funding acquisition:** Myung-Hoon Han.

**Investigation:** Chansub Shin.

**Methodology:** Myung-Hoon Han.

**Resources:** Jin Hwan Cheong.

**Supervision:** Jae Min Kim, Jin Hwan Cheong, Je Il Ryu, Yu Deok Won, Yong Ko.

**Visualization:** Myung-Hoon Han.

**Writing – original draft:** Chansub Shin, Myung-Hoon Han.

**Writing – review & editing:** Je Il Ryu.

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
