## [Decision Letter · Decision Letter 0]

19 Mar 2021

PONE-D-21-00935

Association between tumor size and peritumoral brain edema in patients with convexity and parasagittal meningiomas

PLOS ONE

Dear Dr. Han,

Thank you for submitting your manuscript to PLOS ONE. After careful consideration, we feel that it has merit but does not fully meet PLOS ONE’s publication criteria as it currently stands. Therefore, we invite you to submit a revised version of the manuscript that addresses the points raised during the review process.

We look forward to receiving your revised manuscript.

Kind regards,

Dandan Zheng, PhD

Academic Editor

PLOS ONE

Journal Requirements:

Reviewers' comments:

Reviewer's Responses to Questions

**Comments to the Author**

1. Is the manuscript technically sound, and do the data support the conclusions?

Reviewer #1: Yes

Reviewer #2: Yes

2. Has the statistical analysis been performed appropriately and rigorously? 

Reviewer #1: N/A

Reviewer #2: Yes

3. Have the authors made all data underlying the findings in their manuscript fully available?

Reviewer #1: Yes

Reviewer #2: Yes

4. Is the manuscript presented in an intelligible fashion and written in standard English?

Reviewer #1: Yes

Reviewer #2: Yes

5. Review Comments to the Author

Reviewer #1: Major Comments

1. Has there been strong evidence to show the impact of the tumor locations on the PTBE prediction? The entire study was designed based on the volume of the target, and the spherical shape was not considered as a constraint in the analysis until the volume was converted into diameter in the conclusion.

2. Why would it be necessary to use the approximated spherical diameter as a guideline, instead of using the volume threshold directly? Since the 3D assessment tools are widely available, it may be more accurate to use volume directly for the PTBE prediction.

3. When comparing to the “3cm” cutoff rule of thumb, it’s not clear when this rule of thumb is applied. Is it only applied to spherical meningiomas in practice, or it’s also applied to irregular shape meningiomas in general?

4. It seems Simis, Andre’s study from the reference #3 has concluded a range of PTV diameter where the PTBE is likely to occur. What’s the most distinguished aspect of current study compared to it?

Minor Comments

1. What’s the confidence level of the volume threshold found in this study?

2. How were the two age groups determined?

Reviewer #2: I find the submitted manuscript to be well written, scientifically meaningful, and thoroughly researched and analyzed. The findings of the study are also clinically meaningful as the authors were able to provide a research based explanation for the clinically accepted 3cm diameter cutoff.

6. PLOS authors have the option to publish the peer review history of their article (what does this mean?). If published, this will include your full peer review and any attached files.

Reviewer #1: No

Reviewer #2: **Yes: **Sarah Wisnoskie

---

## [Author Response · Author response to Decision Letter 0]

22 Mar 2021

Response to reviewers 

Association between tumor size and peritumoral brain edema in patients with convexity and parasagittal meningiomas

Chansub Shin, Jae Min Kim, Jin Hwan Cheong, Je Il Ryu, Yu Deok Won, Yong Ko, Myung-Hoon Han

Reviewer #1 

Major Comments

1. Has there been strong evidence to show the impact of the tumor locations on the PTBE prediction? The entire study was designed based on the volume of the target, and the spherical shape was not considered as a constraint in the analysis until the volume was converted into diameter in the conclusion.

Thank you. Previous studies revealed that PTBE occurrence and grade were associated with meningioma location [1–3].

References

 1. Inamura T, Nishio S, Takeshita I, Fujiwara S, Fukui M. Peritumoral Brain Edema in Meningiomas–Influence of Vascular Supply on Its Development. Neurosurgery. 1992;31: 179–185. 

2. Simis A, Pires de Aguiar PH, Leite CC, Santana PA, Rosemberg S, Teixeira MJ. Peritumoral brain edema in benign meningiomas: correlation with clinical, radiologic, and surgical factors and possible role on recurrence. Surg Neurol. 2008;70: 471–477; discussion 477. 

3. Ha Paek S, Kim C-Y, Kim YY, Park IA, Kim MS, Kim DG, et al. Correlation of Clinical and Biological Parameters with Peritumoral Edema in Meningioma. J Neurooncol. 2002;60: 235–245. 

Therefore, we initially only included convexity and parasagittal meningiomas in the study to reduce the possible impact of tumor-location heterogeneity on PTBE occurrence, as described in the Introduction. To address your comment, we added a relevant sentence (including new references) to the Introduction as follows: 

Introduction

However, because meningioma is a tumor with a wide variety of locations, most previous studies that investigated the association between meningioma and PTBE had substantial tumor location heterogeneity. It has been reported that the meningioma location affects PTBE occurrence and grade [3,5,6]. Therefore, we felt it was necessary to include meningiomas with similar locations in this study to reduce the effect of location heterogeneity on the association between meningioma and PTBE. 

As Reviewer #1 indicated, we focused on tumor volume while restricting the heterogeneity of meningioma locations. We agree that the meningioma diameter of 3 cm, which is derived from the volume threshold of 13.953 cc in our study, is solely based on the assumption that the meningioma had a spherical shape. Our premise is that the closer the meningioma shape is to a sphere, the more accurately a diameter of 3 cm can be used for PTBE prediction. In contrast, PTBE prediction for irregularly shaped convexity or parasagittal meningiomas should be based on the direct volume threshold instead of a meningioma diameter of 3 cm. Therefore, we have added relevant sentences to the study limitations described in the Discussion section as follows: 

Discussion

Fourth, not all meningiomas were pathologically confirmed by surgical treatment. Therefore, there may be bias in our results. Fifth, the meningioma diameter of 3 cm derived from the volume threshold of 13.953 cc in our study was solely based on the assumption that it had a spherical shape. Our premise is that the closer the meningioma shape is to a sphere, the more accurately PTBE prediction can be based on a diameter of 3 cm. In contrast, PTBE prediction for irregularly shaped convexity or parasagittal meningiomas should be based on the direct volume threshold instead of a diameter of 3 cm. Lastly, there may be other factors that should be considered in PTBE development in meningiomas, such as tumor location, histological differentiation, hormonal receptors, and arterial tumor supply, which are not covered in our study [31-33].

2. Why would it be necessary to use the approximated spherical diameter as a guideline, instead of using the volume threshold directly? Since the 3D assessment tools are widely available, it may be more accurate to use volume directly for the PTBE prediction.

Thank you, and we agree with your comment. The main purpose of this study as described in the Introduction was to validate the 3 cm diameter meningioma size as a clinically meaningful standard for treatment. Most physicians decide whether to perform surgery or initiate radiation treatment based on a meningioma diameter of 3 cm without any scientific reference. To address this issue, we derived the meningioma diameter from the volume threshold to predict PTBE occurrence. Moreover, we assumed that the meningioma was a complete sphere when deriving its diameter from the volume threshold. As Reviewer #1 suggested, we added relevant sentences regarding the direct volume threshold to the Discussion and Conclusion sections as follows: 

Discussion

Our study showed an approximately 1.2-fold increased risk for PTBE per 1 cc increase in meningioma volume in patients with convexity and parasagittal meningiomas. The overall meningioma volume threshold for predicting PTBE occurrence with high specificity and sensitivity was approximately 14 cc. When meningioma is assumed to be a complete sphere, an approximate 3 cm diameter was the optimal cut-off value for PTBE prediction with high specificity and sensitivity. On the other hand, elderly patients were more vulnerable to both PTBE occurrence and higher grade PTBE for the same or smaller meningioma volume compared to younger patients.

In conclusion, despite these limitations, our study is the first to suggest a cut-off meningioma diameter volume for prediction of PTBE in patients with convexity and parasagittal meningiomas. The meningioma volume threshold for predicting PTBE occurrence was approximately 14 cc. In addition, we believe that we have revealed why meningioma diameter of 3 cm is clinically meaningful.

3. When comparing to the “3cm” cutoff rule of thumb, it’s not clear when this rule of thumb is applied. Is it only applied to spherical meningiomas in practice, or it’s also applied to irregular shape meningiomas in general?

We understand your comment. Meningiomas are usually spherical or lobulated [1].

1. Midyett FA, Mukherji SK. Anterior Fossa Meningioma. In: Midyett FA, Mukherji SK, editors. Skull Base Imaging: The Essentials. Cham: Springer International Publishing; 2020. pp. 81–85. doi:10.1007/978-3-030-46447-9_13

To date, most physicians in clinical practice have measured the diameters of variously shaped meningiomas (usually spherical or lobulated). A diameter of 3 cm has traditionally been considered a clinically meaningful standard for the meningioma treatment. As a result, many neurosurgeons and radiologists typically measure the diameter of variously shaped meningiomas, with the exception of those that are irregularly shaped. As described above, the closer the shape is to a sphere, the more likely a meningioma diameter of 3 cm can effectively be applied to PTBE prediction. PTBE prediction for irregularly shaped meningiomas should be based on the direct volume threshold instead of a 3 cm diameter. Therefore, as Reviewer #1 suggested, we added the following relevant sentences regarding the direct volume threshold to the Discussion and Conclusion sections: 

Discussion

Our study showed an approximately 1.2-fold increased risk for PTBE per 1 cc increase in meningioma volume in patients with convexity and parasagittal meningiomas. The overall meningioma volume threshold for predicting PTBE occurrence with high specificity and sensitivity was approximately 14 cc. When meningioma is assumed to be a complete sphere, an approximate 3 cm diameter was the optimal cut-off value for PTBE prediction with high specificity and sensitivity. On the other hand, elderly patients were more vulnerable to both PTBE occurrence and higher grade PTBE for the same or smaller meningioma volume compared to younger patients.

In conclusion, despite these limitations, our study is the first to suggest a cut-off meningioma diameter volume for prediction of PTBE in patients with convexity and parasagittal meningiomas. The meningioma volume threshold for predicting PTBE occurrence was approximately 14 cc. In addition, we believe that we have revealed why meningioma diameter of 3 cm is clinically meaningful.

In addition, we have added relevant sentences to the limitations in the Discussion section as follows:

Discussion

Fourth, not all meningiomas were pathologically confirmed by surgical treatment. Therefore, there may be bias in our results. Fifth, the meningioma diameter of 3 cm derived from the volume threshold of 13.953 cc in our study was solely based on the assumption that it had a spherical shape. Our premise is that the closer the meningioma shape is to a sphere, the more accurately PTBE prediction can be based on a diameter of 3 cm. In contrast, PTBE prediction for irregularly shaped convexity or parasagittal meningiomas should be based on the direct volume threshold instead of a diameter of 3 cm. Lastly, there may be other factors that should be considered in PTBE development in meningiomas, such as tumor location, histological differentiation, hormonal receptors, and arterial tumor supply, which were not covered in our study [31-33].

4. It seems Simis, Andre’s study from the reference #3 has concluded a range of PTV diameter where the PTBE is likely to occur. What’s the most distinguished aspect of current study compared to it?

As described in the manuscript text and based on clinical experience or implicit consent, most physicians decide whether to perform surgery or initiate radiation treatment based on a meningioma diameter of 3 cm, without any scientific reference. Accordingly, our aim was to validate 3 cm as a clinically meaningful standard for treatment. To the best of our knowledge, this is the first study to suggest an optimal meningioma volume threshold that predicts PTBE in patients with convexity and parasagittal meningiomas. According to our findings, the optimal meningioma volume threshold for prediction of PTBE was 13.953 cc. This finding suggests that a diameter of approximately 3 cm is the optimal cutoff value for PTBE prediction in meningiomas that are assumed to be complete spheres. 

To enhance the accuracy of meningioma volume measurement, we used 3D slicer software. In contrast, Simis et al. measured meningioma volume using the spheroid formula (V = 4/3π × abc) method. In addition, Simis et al. demonstrated that PTBE occurrence or size was associated with large meningiomas. However, in this study we verified that a meningioma diameter of 3 cm is a clinically meaningful standard for treatment. 

Minor Comments

1. What’s the confidence level of the volume threshold found in this study?

Thank you. As Reviewer #1 suggested, we also calculated the 95% confidence intervals in the ROC analyses, as follows: 

We also modified Figure 2 by adding information on 95% confidence intervals as follows:

2. How were the two age groups determined?

We apologize for any confusion. As indicated in the Figure 2 and 3 legends, we defined younger patients as aged <65 years, and older patients as aged ≥65 years. We added relevant sentences to the Statistical Analysis section of the Methods as follows:

Statistical Analysis

Patient data are expressed as mean ± standard deviation or median with interquartile range for continuous variables, and as a count and percentage for discrete variables. The chi-square test and Student’s t-test were conducted to assess differences between the two groups. Younger patients were defined as aged <65 years, and older patients were defined as aged ≥65 years.

---

## [Editor Report · Decision Letter 1]

26 May 2021

Association between tumor size and peritumoral brain edema in patients with convexity and parasagittal meningiomas

PONE-D-21-00935R1

Dear Dr. Han,

We’re pleased to inform you that your manuscript has been judged scientifically suitable for publication and will be formally accepted for publication once it meets all outstanding technical requirements.

Kind regards,

Dandan Zheng, PhD

Academic Editor

PLOS ONE

---

## [Editor Report · Acceptance letter]

2 Jun 2021

PONE-D-21-00935R1 

Association between tumor size and peritumoral brain edema in patients with convexity and parasagittal meningiomas 

Dear Dr. Han:

I'm pleased to inform you that your manuscript has been deemed suitable for publication in PLOS ONE. Congratulations! Your manuscript is now with our production department. 

Kind regards, 

on behalf of

Dr. Dandan Zheng 

Academic Editor

PLOS ONE